# Noise Disturbance and Well-Being in the North of Spain

**DOI:** 10.3390/ijerph192416457

**Published:** 2022-12-08

**Authors:** Maite Santurtún, María José García Tárrago, Pablo Fdez-Arroyabe, María T. Zarrabeitia

**Affiliations:** 1Centro Hospitalario Padre Menni, 39012 Santander, Spain; 2Nursery Department, University of Cantabria, 39005 Santander, Spain; 3Department of Electromechanical Engineering, University of Burgos, 09001 Burgos, Spain; 4Department of Geography, Urban Planning and Territorial Planning, University of Cantabria, 39005 Santander, Spain; 5Unit of Legal Medicine, Department of Physiology and Pharmacology, University of Cantabria—IDIVAL, 39005 Santander, Spain

**Keywords:** noise, well-being, Spain, traffic noise

## Abstract

Environmental noise is considered one of the main risks for physical and mental health and well-being, with a significant associated burden of disease in Europe. This work aims to explore the main sources of noise exposure at home and its effect on well-being in northern Spain. A transversal opinion study has been performed through a closed questionnaire. The questionnaire included three different parts: sociodemographic data, noise disturbance, and the 5-item World Health Organization Well-Being Index (WHO-5). A Binary Logistics Regression model was performed to analyze the relationship between noise exposure and well-being. Overall, 16.6% of the participants consider that the noise isolation of their homes is bad or very bad. The noise generated by the neighbors (air and impact noise) is considered the most disturbing indoor noise source, while street works are the most disturbing outdoor noise source in urban areas and road traffic is the most disturbing in rural areas. People who indicate that noise interferes with their life at home have a worse score on the WHO-5 (decreased perception of well-being). The exposure to outdoor noise (specifically the noise coming from the street and trains), internal impact noise produced by neighbors, and in general, the noise that wakes you up, is related to receiving a worse score in the WHO-5 (*p* < 0.05). Administrative bodies must ensure that laws regulating at-home noise levels, which are continually being updated with stricter restrictions, are enforced.

## 1. Introduction

Environmental noise is defined as any unwanted sound created by human activities that is considered harmful or detrimental to human health and quality of life [1]. Moreover, the World Health Organization (WHO) considers that the environmental noise together with air pollution and water pollution is one of the major forms of environmental pollution, thus, becoming one of the main risks for physical and mental health and well-being with a significant associated burden of disease in Europe [2]. By definition, pollution is something that is to be avoided, controlled, regulated or eliminated [3].

The Environmental Noise Directive (END) 2002/49/EC [4] is the main EU law regarding noise. The aim of the END is to provide a common framework to avoid, prevent or reduce the harmful effects of exposure to environmental noise. The directive requires EU countries to prepare and publish noise maps and noise management action plans [5]. One of the four action areas of the END, which is related to the subject of this investigation, consists of determining exposure to environmental noise and assessing its health effects at the single dwelling level.

The END specifies a number of noise indicators to be applied by Member States in noise mapping and action planning. The most important are L_den_ and L_night_. The L_den_ indicator is an average sound pressure level throughout the day, evening and night, to which a citizen is exposed over a period of a year. L_night_ is the night equivalent level. Both indicators are provided for exposure at the most exposed façade, outdoors.

There are many different sources of environmental noise, for example: transport (road, rail and air traffic), construction and industry, community sources (neighbors, radio, television, bars and restaurants) and social and leisure sources (portable music players, fireworks, toys, rock concerts, firearms, snowmobiles, etc.) [6].

The predominant source of noise annoyance in residential areas is traffic followed by neighbors. Several studies demonstrate that traffic noise causes non-auditory stress effects such as changes in the physiological system, various cognitive deficits, sleep disturbances, modifications of social behavior, psychosocial stress-related symptoms and emotional/motivational effects [7]. Neighbor noises are more complex and generally more difficult to quantify. Usually, neighborhood noise are sounds with high information content such as language, music or also de noise of footsteps. Humans tend to pay attention to these sounds even if the sound level is relatively low. Therefore, the annoyance potential of neighborhood noise is relatively high also at low noise levels. A study carried out in eight cities in Europe [8] concludes that neighbor noise affects health via long-lasting sever annoyance. Moreover, sleep disturbance from neighbor noise is reported as almost on the same level as for traffic, while other noise sources are far below [9].

The World Health Organization (WHO) recommends, through the Environmental Noise Guidelines (ENG), some maximum levels for road traffic noise, railway noise, etc. for the European Union [2]. For instance, the ENG recommends reducing noise levels produced by road traffic below 53 dB L_den_ and 45 dB L_night_ during night time. The ENG also recommends maximum noise levels from railway noise, aircraft noise, wind turbine noise and leisure noise. In addition, the World Health Organization (WHO) Guidelines for Community Noise [6] recommend less than 30 A weighted decibels (dBA) in bedrooms during the night for good sleep quality and less than 35 dBA in classrooms to allow good teaching and learning conditions.

Recently, Perna et al. analyzed the specifications in noise policies and proposed a methodology to compare environmental noise limits [10]. They concluded that some noise policies require a more decentralized approach to adapt to their local contexts. In 2003, Spain introduced their first national noise law [11]. As well as being protected by this, different regions and municipalities have also introduced their own guidelines and regulations. The region of Castilla and León has its own law [12] and the indoor emissions limits are similar to those presented in The Guidelines for Community Noise presented by the WHO. In this case, the maximum recommend noise level in bedrooms is 30 dBA during the night, which concurs with the maximum level in classrooms for good teaching.

Noise is not only a physical stimulus but also an individually experienced noise-event with a corresponding emotional reaction [13,14]. An insufficient ability to cope with noise can therefore lead to an inadequate neuro-endocrine reaction and to regulatory diseases. WHO considers that health is a state of complete physical, mental and social well-being and not merely the absence of disease or infirmity. According to the WHO, well-being is a positive state experienced by individuals and societies. It is a resource for daily life and is determined by social, economic and environmental conditions [15].

In order to be able to implement noise policies with a local context, it is necessary to determine the specific sources of exposure to environmental noise and its impact on the quality of life. Specifically, the objective of this study is to explore the main sources of noise exposure at home and its effect on well-being in northern Spain.

## 2. Materials and Methods

An observational study with a cross-sectional design was performed through a closed questionnaire. Cross-sectional designs are commonly chosen for population-based surveys [16].

Adults (>18 years old) living in Cantabria, a coastland region, and Burgos, an inland region, responded anonymously and voluntarily. These two provinces located in the north of Spain, one coastal and the other inland, were chose due to their different economic characteristics (Burgos is one of the most industrial provinces in Spain, while in Cantabria, this sector is not representative) and in population density (Cantabria has 110 inhab/km^2^, while Burgos has 25.54 inhab/km^2^); these factors are related to noise, and including participants from both regions allows us to analyze different types of noise exposure.

The study was approved by the research projects Ethics Committee of the University of Cantabria.

The questionnaire comprised three different parts: sociodemographic data, noise disturbance and well-being index. Thus, the first part of the questionnaire collected information such as age, sex, income of the household, urban or rural areas. Next, the second part of the questionnaire included questions related with noise disturbance and its impact. This part was an adapted version of the questionnaire employed by Wang and Norbäck [17]. Although the details of the survey can be found in the cited work, it should be noted that 3 groups of questions were included: a first group focused on internal noise sources, the second focused on external noise sources, and finally, a last group focused on specific traffic noise. An adaptation was made about the mentioned survey, which consisted of including new noise sources which are common in the area under study (works in the street, industry, coffee bars, pubs, noise from crowded streets).

Finally, the third part employed the 5-item WHO-Five Well-Being Index (WHO-5), which is a short self-reported measure of current mental well-being over the last two weeks. The 5-item World Health Organization Well-Being Index (WHO-5) is among the most widely used questionnaires for assessing subjective psychological well-being [15]. The WHO-5 consists of five positively worked items that are rated on a 6-point Likert scale, ranging from 0 (none of the time) to 5 (all of the time), see Table 1. Although the total raw score on WHO-5 goes from 0 to 25, for interpretation, its results are multiplied by 4, transforming to a score from 0 to 100. Lowers scores indicating worse well-being. A score of <50 indicates poor well-being and suggests further investigation into possible symptoms of depression [18,19].

In the statistical analysis, a chi-square test was applied to compare the noise disturbance impact depending on the sex and whether the person lived in a rural or urban area. The participants were divided in two groups: those with a Five Well-Being Index (WHO-5) score lower than 50 points (with low mood) and those with a score higher than 50 points. Subsequently, a univariate analysis explored each variable in a data set, separately. The univariate analysis employed the WHO-5 as the dependent variable, while the independent variables were age, economic status, urban or rural living area, noise exposure time and noise impact level. The chi-square test and Mann–Whitney U Test were employed to evaluate the influence of the changes in those variables in the WHO-5 score. The variables with significant influence in the WHO-5 score during the univariate analysis (two-tailed *p* < 0.05) were included in a Binary Logistics Regression model. This model was created for all participants, and also, two different models were independently created for men and women. Finally, the odds ratio and 95% confidence intervals were calculated to assess the relationship between well-being and noise exposure.

## 3. Results

A total of 344 adults participated in the study of whom 316 answered sociodemographic questions; see Appendix A Table A1. Here, 38.3% of the participants were men and 18.7% lived in rural areas. Regarding the score in the well-being index, it should be noted that the differences between sexes were not statistically significant (*p* < 0.05) neither in the specific score nor when comparing people with a score <50 points (poor well-being) with those with a higher score. There were also no statically significant differences when comparing the WHO-5 score segregated by sex and attending to the area of residence; see Table 2.

Overall, 50.5% of the survey respondents considered that the noise isolation of their homes were good or very good, while for 16.6% of the participants, their isolations were bad or very bad. Regarding the perception of noise isolation at home, there were no significant differences if the participant was male or female (*p* > 0.05), but they appear depending on whether the house was located in a rural or urban area (*p* = 0.01); while 29% of the participants living in rural environments considered that the noise isolation was “very good”, only 14.6% of participants in urban areas considered that the noise isolation was “very good”.

During the analysis of the influence of the internal noise sources, the noise generated by the neighbors (due both to air or impact) was considered the most disturbing noise, and differences between rural and urban environments were only significant when the noise was originated in commercial premises inside the building (*p* = 0.019).

As regards external noise sources, the most disturbing noise in urban areas was due to street works, while in rural areas, the noise generated by road traffic was the most referenced (Figure 1) *(*Appendix A Table A2).

Overall, 26.8% of survey respondents were disturbed by traffic noise occasionally, and 3.5% suffer from traffic noise once a week. Moreover, 18.2% of respondents had difficulties falling asleep, and 21.2% were awoken by the noise.

In assessing the relation between well-being level and traffic noise, it was observed that those respondents which were disturbed by traffic noise during at home activities (from watching TV to sleep) had worse score in the WHO-5; see Table 3.

Finally, the study investigates the effects of the different variables depending on whether the WHO-5 is above or below 50 points (as stated above, this score is considered poor well-being) [18,19].

In the univariate analysis, no statistically significant differences were found between obtaining a score above or below 50 points and gender, place of residence (rural vs. urban) and discomfort caused by indoor noise from fans (*p* > 0.05); however, there were significant differences with the rest of the study variables (*p* < 0.05). By introducing all the variables whose differences were significant in the binary logistic regression model (Table 4), we identified that the exposure to external environmental noise (specifically the noise coming from the street and trains), internal impact noise produced by neighbors, and in general, the noise that wakes you up, was related to receiving a worse score on the WHO-5. By making a separate model for each sex, it was found that in males, the relationship was only statistically significant with exterior noises, pubs and street works; while in the case of women, there was a relationship with age, with exposure to external environmental noise (specifically the noise coming from trains) and with the impact noises produced by neighbors.

## 4. Discussion

The main finding of this study is that people who indicate that noise interferes with their life at home have a worse score on the WHO-5 (decreased perception of well-being). In addition, a similar percentage of the population is disturbed by outdoor and indoor noise. Regarding external sources, construction works are what most interfere with the home life of people who live in urban environments, and road traffic is the most unpleasant in rural environments. Internal noise is mainly irritating for people who live in urban environments, especially that caused by neighbors, both airborne (voice, radio, music, among others) and impact-related (scraping sound, footsteps, thumping).

It is remarkable that nowadays, most of the studies focus on the noise caused by traffic [20,21]; however, the impact of construction operations is seldom evaluated, and the results of this research place it as one of the most irritating noise sources.

In our results, we did not find differences between sex and level of well-being, but we detected statistically significant differences in the noise source of disturbance between sexes. Although outdoor noise annoyed the entire population, indoor noise mainly causes discomfort in women. Some authors have described differences in the impact of noise according to sex. Beutel et al. analyzed whether noise annoyance predicts depression, anxiety and sleep disturbance, and they concluded that past noise annoyances were risk factors for mental distress and that women were particularly susceptible to noise annoyance [22]. In this line, Michaud et al. showed that traffic noise annoyance was greater among women [23].

In our results, the score in the level of well-being does not vary significantly depending on whether people live in rural or urban areas, but differences appeared when studying the level of noise disturbance and the main sources of noise.

A study conducted in Canada to investigate expectations and attitudes toward environmental noise in rural and non-rural areas describes that self-reported health status and noise sensitivity were unrelated to geographic areas. However, the prevalence of reporting their area as often or always calm, quiet, and relaxing was 76.8% in rural/remote, 64% in suburban, and 48.4% in urban regions [24].

In 2020, the report published by the European Environment Agency about environmental noise pollution estimated that more than 50% of urban area inhabitants were affected by traffic noise levels of at least 55 decibels in the 2012–2017 period. Moreover, they showed other sources of noise, with the population exposed to railway noise being higher than that exposed to aircraft noise, with those exposed to noise of industrial origin in a distant third position [25]. It should be noted that the previous report analyzed the measured exposure, while the present study is focused on individual perception.

There are hardly any studies on objective measurements and subjective perception of noise. However, its impact on well-being and health seems to be more related to perception than to noise volume [21].

The perception of noise depends on its physical characteristics, as well as on the subject’s mood, attitude, and previous experiences [21], and on individual anatomy and physiology [26]. Moreover, personal circumstances can affect noise perception, which could in turn be related to environmental awareness and education [21,27,28].

Although in recent years, there has been an increase in studies aimed at analyzing the impact of noise in the work environment [29,30], along with an ever-growing number of work-related protection measures, it is imperative that individuals feel comfortable and able to rest in their own homes. In this sense, administrative bodies must ensure that laws regulating at-home noise levels, which are continually being updated with stricter restrictions, are enforced. Although, initially, studies were focused on analyzing the effects of environmental noise in annoyance and sleep disturbance [31], higher risk of suffering physical and psychological disorders, especially in adults [32,33], but also in children [34], have been associates. For this reason, the implementation of restrictive measures is essential in residential areas.

Finally, some limitations of the study must be included. Given the scarce data available about noise disturbances prevalence and variability, as well as the variability of well-being instruments, we did not try to make a priori estimation of sample size. Instead, we used a convenience sample size that was in line with other studies in the literature. However, we consider that a larger sample would strengthen the study. On the other hand, this work focuses on the subjective perception of individuals about noise but does not take objective measurements; this would be interesting to assess the impact on well-being and to be able to establish specific thresholds in the study areas. Finally, the wide dissemination and unrestricted nature of the survey lead to some population groups being underrepresented: for example, there is a higher prevalence of participants from urban areas enrolled in study.

## 5. Conclusions

Noise pollution, one of the great environmental challenges in the world, has a negative impact on people’s lives, both in rural and urban environments. In our study, most participants reported disturbance from external sources of noise in rural and urban areas with effect on their well-being at home. With this context, administrations should implement restrictive measures and more rigorous controls to improve the quality of life of the citizens.

## Figures and Tables

**Figure 1 ijerph-19-16457-f001:**
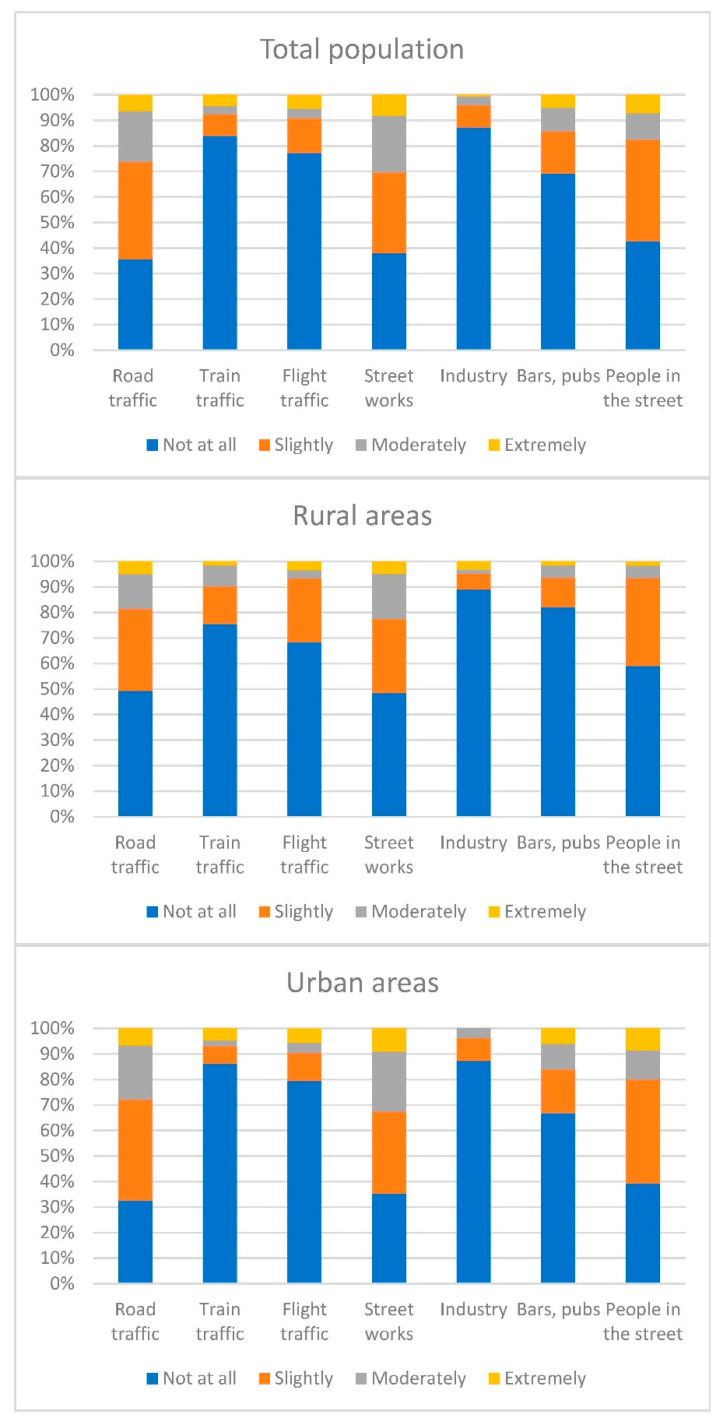
Percentage of participants who reported disturbance from external sources of noise in rural and urban areas during the previous 3 months.

**Table 1 ijerph-19-16457-t001:** WHO-Five Well-Being Index (WHO-5).

Please Respond to Each Item by Marking One Box per Row, Regarding How You Felt in the Last Two Weeks.		All of the Time (5)	Most of the Time (4)	More Than Half the Time (3)	Less Than Half the Time (2)	Some of the Time (1)	At No Time (0)
WHO-1	I have felt cheerful in good spirits.	◯	◯	◯	◯	◯	◯
WHO-2	I have felt calm and relaxed.	◯	◯	◯	◯	◯	◯
WHO-3	I have felt active and vigorous.	◯	◯	◯	◯	◯	◯
WHO-4	I woke up feeling fresh and rested.	◯	◯	◯	◯	◯	◯
WHO-5	My daily life has been filled with things that interest me.	◯	◯	◯	◯	◯	◯

**Table 2 ijerph-19-16457-t002:** Demographic characteristics and living environment of the participants.

	Total		Men		Women	
	Rural	Urban	Rural	Urban	Rural	Urban
18–44 years	37	140	18	56	19	84
45–64 years	14	56	1	18	13	38
65+ years	8	61	5	23	3	38

**Table 3 ijerph-19-16457-t003:** Relationship between well-being level and the disturbing effect of traffic noise in the life at home. The table shows participant’s average score, standard deviation, maximum and minimum punctuation in the Five Well-Being Index (WHO-5). For the interpretation, it should be taken into account that a score of 0 represents worst thinkable well-being, and 100 is the best thinkable well-being.

	Never				Yes, Sometimes				Yes, Often			
	Average	SD	Max.	Min.	Average	SD	Max.	Min.	Average	SD	Max.	Min.
Difficult to hear radio/TV	64.5	18.3	100.0	12.0	57.8	19.9	100.0	20.0	47.0	21.5	80.0	28.0
Telephone calls being affected	64.6	18.0	100.0	12.0	52.1	20.4	100.0	20.0	42.5	16.8	72.0	28.0
Conversations at home being affected	64.3	18.4	100.0	12.0	51.8	19.5	84.0	20.0	42.0	14.7	64.0	28.0
Rest/relaxation being disturbed	66.5	18.2	100.0	12.0	54.5	18.0	96.0	16.0	48.7	17.6	80.0	32.0
Difficulties in sleeping	66.1	17.9	100.0	12.0	50.5	17.6	88.0	16.0	47.0	18.3	80.0	28.0
Being woken up	65.6	18.1	100.0	12.0	53.7	18.9	96.0	16.0	52.0	20.0	80.0	32.0

**Table 4 ijerph-19-16457-t004:** Results of the Binary Logistics Regression model. The dependent variable was the Five Well-Being Index (WHO-5) categories as: lower than 50 points (low mood) or higher than or equal to 50 points (normal mood); the independent variables are age, economic status, urban or rural living area, noise exposure time and noise impact level. Only statistically significant variables (*p* < 0.05) are included.

	Variables		OR (95% CI)	*p* Value
**Total**	Age		0.960 (0.939–0.981)	<0.001
	Noise from sources inside the building	Scraping sound/footsteps/thumping/similar sounds from neighbors	1.647 (1.142–2.375)	0.008
	Noise from sources outside the building	Disturbed by train traffic	2.234 (1.136–4.394)	0.02
		Disturbed by outside street/plaza noise	1.846 (1.236–2.758)	0.003
	Effect of noise	Difficulties in sleeping	2.184 (1.137–4.194)	0.019
**Men**	Disturbed by noise from sources outside the building	Street/construction works	1.870 (1.128–3.099)	0.015
		Bars, pubs	2.104 (1.183–3.741)	0.011
**Women**	Age		0.951 (0.924–0.980)	0.001
	Disturbed by noise from sources inside the building	Impact noises: footsteps/knocks/others	1.987 (1.234–3.199)	0.005
	Disturbed by noise from sources outside the building	Train traffic	3.343 (1.209–9.245)	0.02

## Data Availability

The data that support the findings of this study are available from the corresponding author upon reasonable request.

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
