# Peer review of "Noise Disturbance and Well-Being in the North of Spain"

_ijerph, 2022, doi:10.3390/ijerph192416457_

Round 1

Reviewer 1 Report

The topic under study is very interesting for the audience. However, in my opinion major revisions should be provided by Authors before considering it for publication. I list my suggestions hereafter.

Introduction: the Authors should implement relevant publications on the topic from all the world, Europe, and Spain.

Methods: Authors should give further details of the second and third parts of the questionnaire, to better understand results shown in Tables 1 and 2. For example, answers in WHO-5 is written to be given on a 6-point Likert scale, but in Table 2 only three columns are reported; further, six items are reported in Table 2 while in the Methods section is written that the WHO-5 has five items (?). All the reported variables in Results should be included and well-explained in the Methods section. 

Results: the Authors should revise the row percentages shown in Table 1: they do not seem to match. Tables should also be edited accordin to guidelines for authors. Authors should report results from univariate analysis at least in the text, to ascertain which variables were then included in the logistic regression model. 

Discussion: Authors should provide an extensive improvement of the section, inclusing a proper discussion of all findings (e.g., comparison between urban and rural area, and between men and women; limitations of the study related for instance to the higher prevalence of participants from urban areas enrolled in study).

Finally, Authors should revise some typos found in the text (e.g. "test" reapeated twice at line 84, "focused in" in place of "focused on" at line 154).

Author Response

Reviewer 1

The topic under study is very interesting for the audience. However, in my opinion major revisions should be provided by Authors before considering it for publication. I list my suggestions hereafter.

Introduction: the Authors should implement relevant publications on the topic from all the world, Europe, and Spain.”

Thank you very much for the detailed review. We have extended the introduction and included new references.

“Methods: Authors should give further details of the second and third parts of the questionnaire, to better understand results shown in Tables 1 and 2. For example, answers in WHO-5 is written to be given on a 6-point Likert scale, but in Table 2 only three columns are reported; further, six items are reported in Table 2 while in the Methods section is written that the WHO-5 has five items (?). All the reported variables in Results should be included and well-explained in the Methods section. “

We have extended the explanation about the questionnaire in methods and results. We have included a new table to explain the well-being index. Table 1 has been replaced by a new figure, “Figure 1”. In table 2, we crossed the questions about noise with the score obtained in the well-being index. It has been detailed in the results and in the legend of the table.

Results: the Authors should revise the row percentages shown in Table 1: they do not seem to match. Tables should also be edited accordin to guidelines for authors. Authors should report results from univariate analysis at least in the text, to ascertain which variables were then included in the logistic regression model. 

We have changed Table 1 for a new Figure which is more convenient. We have report results for univariate analyses in the test.

Discussion: Authors should provide an extensive improvement of the section, inclusing a proper discussion of all findings (e.g., comparison between urban and rural area, and between men and women; limitations of the study related for instance to the higher prevalence of participants from urban areas enrolled in study).

We agree with you. We have improved this section and we have included the limitations of the study.

Finally, Authors should revise some typos found in the text (e.g. "test" repeated twice at line 84, "focused in" in place of "focused on" at line 154).”

Thanks. We have revised the whole manuscript and we have requested an extensive English revision to Journal’s editing service.

Reviewer 2 Report

Abstract :

Please can you give one example of environmental noise

Introduction

Introduction Please reword introduction with more research and more detail about environmental noise: what is environmental noise, definitions? Its consequences? why is it so important to consider it. Please define well-bein and link with noise disturbance.

Please develop your introduction, present the research about impact of noise on well-being. Please add more research reference in our work.

"Recently, Perna et al. analyzed the specifications in noise policies and proposed a methodology to compare environmental noise limits (4). Please develop environmental noise limits ans related results.

Material and methode*

A transversal opinion study: why do you choose this method. Please argument.

Two regions located in the north of Spain: why do you choose thise region? Please argument. Argument your sample method. 

The 78 participants were divided in two groups: those of Five Well- Being Index (WHO-5) lower 79 than 50 points (with low mood), and those of higher than 50 points. Why do you do that? Explain why do you divide group? Is it a theoritical practice? justify!

Results:

Difference between man and woman, please indicate the used analysis in your study. IS it ANOVA? other?

General:

Please complete your work with more references

be careful conjugation, use preterit for results presentation.

Author Response

Reviewer 2

Introduction

Introduction Please reword introduction with more research and more detail about environmental noise: what is environmental noise, definitions? Its consequences? why is it so important to consider it. Please define well-bein and link with noise disturbance.

Please develop your introduction, present the research about impact of noise on well-being. Please add more research reference in our work.

"Recently, Perna et al. analyzed the specifications in noise policies and proposed a methodology to compare environmental noise limits (4). Please develop environmental noise limits ans related results.

We greatly appreciate your comments and agree with them. we have intensively revised the whole introduction.

 Material and methode*

A transversal opinion study: why do you choose this method. Please argument.

Two regions located in the north of Spain: why do you choose thise region? Please argument. Argument your sample method. 

The 78 participants were divided in two groups: those of Five Well- Being Index (WHO-5) lower 79 than 50 points (with low mood), and those of higher than 50 points. Why do you do that? Explain why do you divide group? Is it a theoritical practice? justify!

Thanks. We have clarified these aspects in the methods section.

Results:

Difference between man and woman, please indicate the used analysis in your study. IS it ANOVA? other?

Included under Table 1.

General:

Please complete your work with more references
be careful conjugation, use preterit for results presentation.

Thanks. We have revised the whole manuscript and we have requested an extensive English revision to Journal’s editing service.

Reviewer 3 Report

Please consider additional keywords that can better describe the subject of the article - e.g. the words "outdoor, indoor" will not mean anything to the reader - they need to be related to noise.

Please provide short additional information on why the respondents were divided according to the 50-point criterion (article lines from number 78).

In the article, especially in Chapter 2, there is no reference to the noise limit values. Is it possible to explain why such a reference is not made, but only to the Index WHO-5 and the like. The limit values, especially on roads, can clearly identify problems with sleep disturbance or other indicators.

The authors should try to determine the required size of the sample from the statistical point of view - the number of people subjected to the study.

I propose to show the results of the tables in the form of charts where possible. Such a presentation will show situations and trends much better. This mainly applies to Table 1.

Please rethink some conclusions. Table 1 shows that roads and streets are the source of the worst disturbance. This result is part of the general information on noise pollution.

In Chapter 5 (Conclusions), the thought regarding the implementation of certain measures should be developed. Do the authors mean administrative, technical etc. activities?

Author Response

Reviewer 3

Please consider additional keywords that can better describe the subject of the article - e.g. the words "outdoor, indoor" will not mean anything to the reader - they need to be related to noise.

Thanks. We have included new keywords.

Please provide short additional information on why the respondents were divided according to the 50-point criterion (article lines from number 78).

We have included additional information and references to address this issue.

In the article, especially in Chapter 2, there is no reference to the noise limit values. Is it possible to explain why such a reference is not made, but only to the Index WHO-5 and the like. The limit values, especially on roads, can clearly identify problems with sleep disturbance or other indicators.

In the article we only collected the subjective sensation of noise, we did not make measurements. We have included this aspect in limitations. We would like to include this variable in future studies.

The authors should try to determine the required size of the sample from the statistical point of view - the number of people subjected to the study.

Given the scarce data available about noise disturbances prevalence and variability, as well as the variability of well-being instruments,  we did not try to make a priori estimation of sample size. Instead, we used a convenience sample size that was in line with other studies about wellbeing (10.1007/BF03035123 or 10.1159/000289073 ). However, we have included this issue it in the limitations of the study.

I propose to show the results of the tables in the form of charts where possible. Such a presentation will show situations and trends much better. This mainly applies to Table 1.

 Thanks. We agree with you. We have included a new Figure in the revised version.

Please rethink some conclusions. Table 1 shows that roads and streets are the source of the worst disturbance. This result is part of the general information on noise pollution. In Chapter 5 (Conclusions), the thought regarding the implementation of certain measures should be developed. Do the authors mean administrative, technical etc. activities?

We thank the reviewers for their constructive criticism. We also feel that the manuscript has been improved following their suggestions, and hope it can be now regarded as suitable for publication in the International Journal of Environmental Research and Public Health.

Kind regards.

Round 2

Reviewer 1 Report

I appreciated the authors' work. I suggest they fix some other points I noticed.

Introduction: lines 95-97 could be moved to the Methods section.

Results: In my opinion, the previous Table 1 was clearer than the current Figure 1; therefore, I suggest rewriting it appropriately. Rows 201-202 should be the notes in Table 2 or, alternatively, explained in the Methods section.

In general, in my opinion, a revision of the entire manuscript is still necessary, provided some errors are present. For example, one study is cited in lines 67-69 but two references have been added. Therefore, I urge authors to strictly follow the guidelines for authors (e.g., in-text citations are assumed to be in square brackets; tables should be placed immediately after their in-text citation sentence; editing of tables is also necessary). Typos should also be checked carefully (for example, "mentioned" on line 127 instead of "mentioned"). 

Author Response

We greatly appreciate your comments and agree with them. We hace revised the manuscript according to the new suggestions. All changes are in red in the revised version

Reviewer 3 Report

Thank you for the additions. Good luck.

Author Response

(The authors gave the same response as above.)
